# Structural Performance Improvement of Reinforced Concrete Beams by Strain-Hardening Cementitious Composite Layers

**DOI:** 10.3390/ma16145135

**Published:** 2023-07-21

**Authors:** M. Iqbal Khan, Sardar Umer Sial, Yassir M. Abbas, Galal Fares

**Affiliations:** Department of Civil Engineering, College of Engineering, King Saud University, P.O. Box 800, Riyadh 11421, Saudi Arabia; sardar.umer.sial@gmail.com (S.U.S.); yabbas@ksu.edu.sa (Y.M.A.); galfares@ksu.edu.sa (G.F.)

**Keywords:** strain hardening cementitious composites, composite beams, flexural performance, ductility analysis, cracking response, theoretical prediction

## Abstract

A strain-hardening cementitious composite (SHCC) is a modern engineered material offering exceptional ductility and durability. A potential application of SHCCs for crack control and to improve structural members’ load-bearing capabilities is due to its superior properties. In this study, SHCCs were used to enhance the load-carrying capacity and the cracking behavior of precast RC beams. In the bottom tension region of RC beams, the SHCCs of different layer thicknesses (0%, 15%, 30%, and 45% of section height) were cast. Laboratory-scale beams were used in 4-point bending tests. SHCC-layered RC beams showed improved flexural performance compared to control RC beams. Among retrofitted beams, the one with an SHCC layer of 30% of the section height was found to be the most efficient pertaining to strength, ductility, and cracking control. In this study, the flexural response of composite beams was also predicted using an analytical approach. The average difference between predicted and measured moment capacities was less than 10%.

## 1. Introduction

Concrete deterioration due to overloading and durability issues is a serious concern for the construction sector. The rehabilitation of concrete structures damaged as a result of crack formation in concrete is unavoidable, and cracks of large widths cause durability issues in reinforced concrete (RC) structures due to the infusion of hazardous external agents that initiate the corrosion of steel bars. With the advent of fiber RC, various advanced cementitious materials and composites with improved ductility and durability characteristics have been developed [1,2,3,4,5]. Additionally, several pozzolanic binders, such as metakaolin, fly ash, slag, rice husk ash, and palm oil fuel ash, have been developed to reduce greenhouse gas emissions in the construction industry [6,7,8]. These pozzolanic materials are commonly used as supplementary cementitious materials (SCMs). Habert et al. [9] report that cement accounts for 74–80% of all carbon dioxide emissions from concrete, even though it represents approximately 10% of its mass. A reduction in CO_2_ emissions per unit of concrete can be achieved by replacing parts of the concrete cement with SCMs. A study showed that using SCMs could reduce cement’s CO_2_ emissions by up to 20% if the construction industry adopted them [10]. Due to their often by-product or waste status, SCMs produce significantly less CO_2_ than cement due to their manufacturing processes and by-products. Since most SCM emissions come from preparation and are generally low, they are often overlooked. The use of SCMs made from waste materials not only reduces landfill waste, but these products have a positive effect on the environment as well [11]. 

Recent developments have led to innovative strain-hardening cementitious composites (SHCCs) with superior ductility and multiple-cracking behaviors [12,13,14]. The micromechanics theory is often used to model fibers, matrix, and their interactions in the SHCC design process [5,15,16,17]. Unlike concrete, SHCCs do not experience localized failures, since fibers control crack formation through the bridging effect, resulting in a much more deformable material. The strain capacity and tensile strength of SHCCs incorporating about 2.0% (vol.) fibers range from 2 to 3% and 4 to 6 MPa, respectively [15,18,19,20]. Owing to its superior durability, SHCCs were found to be a promising construction material for different environmental conditions [21,22,23]. The use of SHCCs in construction can be an effective way to control crack formation. In the literature, the feasibility of SHCCs in seismic and non-seismic applications has been examined [24,25,26,27]. One potential application is the development of composite RC-SHCC beams with improved flexural and durability performance. If an SHCC is used in the tensile region of a beam, it can increase the moment-carrying capacity along with the durability of the beam [19,28,29]. A similar improvement has been found in high-strength RC beams with a sandwiched layer of SHCCs in the tensile region [30]. A study [31] indicates that RC beams strengthened with SHCCs and high-strength steel bars have better load-carrying capacities and smaller crack widths. In one numerical study on RC–SHCC composite beams [32]. In the study, the strengthening layer of the SHCC was found to be effective in reducing the stress concentrations in the rebars and the crack widths of the beam. Additionally, the following paragraphs provide an overview of the most recent studies that exploit SHCCs for structural applications. The impact of smooth and rough trapping of kinks and cracks through surface preparation in SHCC-concrete retrofitting is investigated [33]. According to their results, smooth surfaces display superior cracking behavior, which is crucial for constructing durable repairs. Furthermore, the smooth surface treatment effectively prevented the delamination mode of failure associated with kink-crack overlays. Further, Hussein and Nakamura [34] reported improved ductility of RC beams retrofitted with an SHCC layer containing 0.3 and 0.6% steel bars. As a result of applying a bar-reinforced SHCC layer in beam strengthening, premature strain localization was mitigated. Crack behavior was enhanced, resulting in beams displaying ductile failure instead of brittle failure. The strength enhancement ability of steel fiber-reinforced self-compacting concrete (SFRSCC) and SHCC layers during flexural loads on masonry beams was investigated by Esmaeeli et al. [35]. In their study, the SHCC layer improved beam ductility and load capacity more effectively than the SFRSCC layer. Additionally, Afefy and Mahmoud [36] designed precast SHCC strips incorporating small amounts of steel bars as tension reinforcement for one-way RC slabs. Their experiments revealed that slabs containing SHCC strips performed better (in terms of deformability, cracking response, and load strength) than beams without the strips, especially those with a steel bar reinforcement ratio of 1.88%. Moreover, the characteristic brittle failure mode of compression-controlled RC beams was improved by ultrahigh-performance SHCC (UHP-SHCC) retrofitting in Atta and Khalil [37]. It was possible to improve the structural characteristics of beams with a high loading capacity and ductility with high brittleness using the proposed strengthening technique. Moreover, Khalil et al. [38] investigated the response of reinforced RC beams with UHP-SHCC layers to monotonic and cyclic loading, focusing on the layer’s bar-reinforcement ratio. According to their conclusions, a beam with an unreinforced layer is predicted to fail catastrophically under cyclic loading. However, the beam that has a reinforced layer would be ductile and less likely to exhibit localized strains. An investigation of the behavior of polyvinyl alcohol (PVA) and polypropylene fiber (PP)-based bar-reinforced ECC–concrete beams was carried out by Shanour et al. [39]. In comparison with PP fibers, PVA fibers were shown to be more effective at increasing the load-carrying capacity.

A study [40] by Qiao et al. developed a cost-effective and durable permanent framework of U-shaped ECC for RC beams. ECC–RC beams were proven to have better structural behavior than the reference beam. Additionally, Shao and Billington [41] analyzed two failure modes in RC-reinforced SHCC beams (crack localization and steady hardening of reinforcing steel) contributing to ductility and load-carrying capacity concerns. In their study, reinforcement percentage and type were emphasized. In this study, it was demonstrated that reinforcement percentage and type influenced failure mode. Recently, Khan et al. [42] investigated the impact of the hot and steam curing regimes on the flexural properties of RC beams retrofitted with a 50 mm-thick SHCC layer in the tension zone. A lower ductility was observed in the strengthened beams under hot and steam curing. However, steam curing was more effective than hot weather curing for increasing load-carrying capacity. RC beams are one of the most significant structural members. The region below the neutral axis of the RC flexural member is highly critical due to the tensile weakness of concrete. Cracks in the bottom tensile zone of RC beams are inevitable and occur at widths that can seriously damage the structural health of the members. Partially replacing the concrete with a special material like the SHCC layer in the tensile region can provide an extra tensile contribution to restrict the crack widths. In precast plants with a controlled environment, such RC–SHCC composite flexural members can be produced. These beams can have specialized applications in industrial and coastal areas. In light of the above literature review, further research is still required to provide a thorough understanding of the flexural behavior of composite beams made up of RC and SHCCs. The role of the SHCC layer will be covered in the current investigation. The effective thickness of the SHCC layer in the composite beam has not yet been determined for optimal flexural and cracking performance. In cases where precast structures are exposed to harsh weather conditions, where cracks on the surface negatively impact their structural integrity and durability, replacing normal concrete with SHCC could be a viable solution. In this research, the flexural performance of RC–SHCC composite beams with different SHCC replacement levels reaching up to half the section depth was investigated. The thickness of the SHCC layer was the main variable. A detailed experimental analysis of the four-point loading test has been conducted to achieve this goal. The rational analytical prediction has also been developed based on constitutive material and sectional models, which facilitates the application of the strengthening technique.

## 2. Materials 

The binder used in this work is composed of ordinary Portland cement (OPC) and class F fly ash (FA). In this study, an optimized SHCC mix with a water-binder ratio of 26% and a PVA content of 2% (vol-based) was designed and produced. The chemical analysis of OPC and FA is shown in Table 1. The constituent material weights of the formulated SHCC are summarized in Table 1. Table 2 displays the physicomechanical properties of the PVA fibers used. PVA strands were investigated using the Versa-3D dual-beam field emission scanning electron microscope from FEI, as presented in Figure 1. PVA stands have consistent dimensions. The mechanical properties of PVA are documented in Table 2. Both OPC and FA had specific gravities of 3.14 and 2.3, respectively. The Blaine air-permeability device determined their fineness as 3730 and 4500 cm^2^/g, respectively. As for particle size distribution, the median size of OPC was 14 µm, and that of FA was 10 µm. As part of the design of the SHCC, fine silica sand with a size of 200 m was incorporated. This sand passed the 300 m sieve size to ensure satisfactory volume stability. A modified polycarboxylic ether-based superplasticizer (SP) was added to the mix to improve its workability. The dry extract and unit weight of the SP were 36% and 1100 kg/m^3^, respectively.

Table 3 presents the mixture proportions of the normal concrete (NC) that was cast on the beam’s substrate. A target 28-day compressive strength of 35 MPa was specified for the NC. This strength was achieved by a concrete mix made with a water-to-cement ratio of 0.5. The fine aggregates were a mixture of crushed sands and natural sands (65 and 35%, respectively) with an overall modulus of 2.58. The coarse aggregates had a nominal size of 10 mm. A plasticizer was used for the target slump value of above 100 mm. The sieve analysis of aggregates is provided in Figure 2.

The mix composition of normal concrete is provided in Table 4. The normal ingredients of fine and coarse aggregates are included. A cement content of 350 kg/m^3^ with a water-to-cement ratio of 0.5 is used.

## 3. Methods

In the present study, the 28-day compressive strength of substrate concrete specimens measuring 100 mm (D) and 200 mm (L) was evaluated in accordance with ASTM-C39 specifications. The tensile strength of the SHCC was determined by uniaxial tension testing of dumbbell specimens (Figure 3a) using a 30 kN universal testing machine (known as the Instron 3367) under strain-controlled conditions (at 0.15 mm/min displacement rate). In this test, the gauge length was 80 mm, and the displacement response was measured with an extensometer attached to the middle of the test sample. The flexural strength (Figure 3b) was also evaluated by testing the SHCC prisms (75 × 75 × 300 mm) at a displacement rate of 0.2 mm/min. The specimen deflection was detected using two linear variable differential transducers (LVDTs) placed at the mid-span. In this study, we examined the material properties of three typical specimens and presented the average values. 

The research examined RC beams measuring 1800 mm (L) × 200 mm (D) × 150 mm (W) under four-point loading. These beams were characterized by the layer thickness of the SHCCs (0, 30, 60, and 90 mm). In total, these thicknesses account for 0%, 15%, 30%, and 45% of the overall depth. The overall depth of these beams remained constant at 200 mm. In place of the NC, the SHCC was placed at the bottom of the beam instead.

Figure 4 displays a schematic representation of the tested beams. There were two 16 mm steel bars placed at the bottom and top of each beam. The RC beams were reinforced with two-leg, 8 mm-diameter stirrups positioned at 100 mm intervals in the shear zone in order to study their flexural behavior (i.e., to avoid brittle failure in the shear zone). Further, Figure 5 shows the sections of the tested beams. As shown in this figure, the RC beam with the x mm thick SHCC layer is labeled “RC-L-SHCC-x” and the control RC beam is labeled “RC-C”.

Casting was carried out from bottom to top on the specimen mold for the RC-C beam. In Figure 6, the casting procedure for RC beams containing SHCC layers is illustrated. It was necessary to reverse the direction of casting in this case. As a result, the NC was initially cast in a steel cage that had been reversed for the intended height. Wire brushes were used to coarsen the surface of the concrete after setting (to ensure the NC and SHCC layers would bond properly). The final step was to apply the SHCC layer over the rough NC surface. Under standard moist conditions, the RC beams were cured until the testing date (28 days). 

Figure 7 shows the detailed setup for the four-point test of the RC beams. In this test, specimens were subjected to two monotonous loads distributed over 550 mm (see Figure 4) by a hydraulic testing machine (AMSLER with 3000 kN capacity and a servo-controller). Additionally, Figure 7 shows the system of accessories used for obtaining data (strain gauges and LVDTs). The beam’s curvature was evaluated using two LVDTs mounted horizontally beneath and directly above it, while its deflection was measured using one LVDT located vertically at its mid-span. Moreover, the beam’s top surface was fitted with two strain gauges to measure its maximum compressive strain. In addition, the steel bars were equipped with two strain gauges in the middle. For monitoring variations in the sectional strain, there were two more strain gauges glued to the lateral face of the beam. Additionally, a system of high-resolution digital cameras and a crack detection microscope with an X35 magnification, measuring ranges up to 4 mm, 0.02 mm divisions, and a dimension of 40 × 90 × 150 mm were used in order to monitor cracks developing at multiple locations along the beam’s span.

## 4. Results and Discussion

### 4.1. Material Properties

In this study, three typical tension steel bars (with a diameter of 16 mm) were directly tested in tension to examine their key mechanical properties. According to the test results, the employed steel had an average yield strength of 480 MPa and an ultimate strength of 600 MPa, while its elasticity modulus was 201.8 GPa. Meanwhile, the NC showed compressive strength and Young’s modulus of 41 MPa and 22 GPa, respectively, after 28 days. In the case of the SHCC mix, an average compressive strength of 58 MPa was found after 28 days. Figure 8 shows the stress–strain behavior of the SHCC as determined by three dumbbell-shaped specimens tested under uniaxial tension. Additional details regarding this test are provided in [43,44,45], a collection of the authors’ previous research studies. According to the results of this study, the composite’s mean tensile strength was 4.35 ± 0.21 MPa and its strain capacity was 2.0%. The relationship between the midspan deflection and flexural stress is also shown in Figure 9. Therefore, the material had a mean 28-day flexural strength of 20.46 MPa and a mid-span deflection at peak stress of about 2.0 mm. The SHCC mechanical properties shown in Figure 8 and Figure 9 vary within the lower limit of the reported variation range of 10 to 25% [38,39]. 

The SHCC displayed multiple cracking responses (with small crack widths (less than 0.1 mm)) to tensile and flexural loads, as shown in Figure 8 and Figure 9. In Figure 6, a fine cracking pattern can also be seen throughout the sample. A similar pattern emerged under flexural loading, although the pattern diffused exclusively beneath the applied load (Figure 7). The uniform PVA fiber distribution is confirmed through the SEM analysis of many specimens, as typically demonstrated in Figure 10. This uniform distribution is enhanced by the improved workability.

### 4.2. Flexural Behavior of the RC Beams 

#### 4.2.1. Load–Deflection Responses

Figure 11 displays the load–deflection curves of the tested RC beams, while Table 5 summarizes their critical values (related to elasticity limit, yielding, and ultimate phases). Also, the values of loads and deflections of layered beams were compared to those of the control beam, and their differences are shown between brackets as percentages in Table 5. The stiffness of all the beams was similar before the first cracking, but it increased for layered beams in the region between the first cracking and the yielding as compared to the control beam. The yield and ultimate loads of the layered beams are larger than those of the control beam, with a maximum increase of 30.8% and 9% in the RC-L-SHCC-60 beam, respectively. It should be noted that the percentage difference in the load-carrying capacities of the layered beams was not very significant. In terms of deflection, layered beams showed an improvement (10–20%) at the yielding stage but degradation (20–35%) at the ultimate stage as compared to the control beam. 

Among the layered beams, the RC-L-SHCC-60 beam showed maximum deflection values at failure than the others. This increase in the deflection of the layered beams is attributed to the tensile contribution of the SHCC layer [46]. The tensile resistance provided by the SHCC layer and steel rebars balances the compression resistance of the concrete, causing a decrease in the maximum tensile strain of steel rebars. This decrease in steel strain, in turn, decreases the ultimate deflection of the layered beam. Therefore, the RC–SHCC-layered beams have an increased ultimate load capacity (7.5% on average) but a decreased ultimate deflection (27% on average) than that of the control RC beam. Since the RC-L-SHCC-60 beam showed maximum load and deflection capacities as compared to other layered beams, it can be said that the SHCC layer of 30% of section height performs efficiently in composite beams.

#### 4.2.2. Load–Strain Responses

The load–strain relationship of beam specimens is given in Figure 12. It is noted here that strain was measured at the top of the section and bottom rebars in the mid-span of the beam. All the curves have a common trend of an initial linear part followed by a nonlinear part. As the steel yields, the concrete crushes at the top, indicating ductile behavior in the beams. The layered beams are stiffer than the control beam because of their greater elastic stiffness. It can be verified that the tensile strain values in the layered beams have decreased as compared to the control beam, resulting in a loss of the deflection of the layered beams. 

The steel-yielding load increased in the layered beams as compared to the control beam. After yielding, the load became stagnant, and the strains in the steel increased dramatically. Correspondingly, the strain in the compressive concrete at the top also increased until the ultimate crushing strain of the concrete reached the top of the beams. The final load at which the concrete crushed increased in the RC–SHCC-layered beams as compared to the control beam.

#### 4.2.3. Moment–Curvature Responses

The moment–curvature behavior of the tested RC beams is illustrated in Figure 13. A comparison of Figure 11 and Figure 13 shows a similar trend in the moment–curvature and load–deflection behavior. Moreover, Table 6 summarizes their critical moment–curvature values (for the yielding, ultimate, and elasticity limit). It is noted here that the curvature was calculated by dividing strain (at the top face) with the neutral axis depth of the beam. As loads increase, the moments of the layered beams are notably higher than those of the control beam, with a maximum increase in the beam of RC-L-SHCC-60.

On the contrary, the curvatures of the layered beams are less than those of the control beam. At the yield stage, this reduction in curvature was in the range of 5–8%, which was raised to 15–40% at the ultimate stage. As explained earlier, this decrease in the curvature of the layered beams is due to the composite action of the SHCC layer and bottom rebars in resisting tensile stresses. Among the layered beams, the RC-L-SHCC-60 beam showed maximum curvature as compared to the rest. So, it can be restated that the maximum moment capacity and curvature are achieved by a layered specimen with an SHCC layer with a beam depth of 30% thickness.

#### 4.2.4. Ductility Analysis

In flexural members, ductility is crucial, as it indicates how much of the ultimate load it can support while deforming. It has been proven that RC–SHCC-layered beams display reduced deflection/curvature compared to the control RC beams, as discussed previously. This study extensively examined the ductility of SHCC-layered beams to determine the RC beam performance. Thus, ductility indexes were calculated in order to measure the ductility of beams. These parameters were the member ductility (µ_mem_) and section ductility (µ_sec_). These indices are calculated by the ratios of ultimate deflection (δ_u_) to yield deflection (δ_y_) and ultimate curvature (ϕ_u_) to yield curvature (ϕ_y_), respectively. In Table 7, the ductility indices of the beam specimens are given, with their percentage differences from the control specimen in brackets. It can be seen that the ductility indices of layered beams are lower than those of the control beam, with a minimum reduction (8% in µ_sec_ and 27% in µ_mem_) for RC-L-SHCC-60. This loss in ductility occurred due to an additional tension-carrying area (SHCC layer); however, the layered beams still showed promising ductility values. According to the FIP recommendations [47], a minimum section ductility (µ_sec_) of 1.7 for a normal-strength concrete beam should provide a sufficient warning before failure. The section ductility (µ_sec_) values of the layered beams exceed this value and, thus, ensure adequate ductility. Therefore, it can be said that the composite beam RC-L-SHCC-60 with a 60 mm SHCC layer (30% of section height) showed maximum ductility compared to the rest of the layered beams. 

#### 4.2.5. Cracking Patterns

The cracking patterns at the mid-span of beams at the peak load stage are shown in Figure 14. For control specimen ‘RC-C’, the first crack appeared in the mid-span at approximately 15 kN of load (13.3% of the ultimate load). As the load increased, the flexural cracks appeared along the length of the beam and propagated towards the neutral axis of the beam. It was evident that a significant increase in crack width occurred after the steel yielded. At the peak load, the crushing of the concrete occurred at the mid-span of the beam. For beam ‘RC-L-SHCC-30′, the first crack initiated in the mid-span at a 20 kN load (16.3% of the peak load). 

The SHCC layer and concrete substrate were more likely to fracture with increasing loads. A comparison of SHCC cracks with NC cracks revealed that the SHCC cracks were much finer. In contrast to the SHCC layer, the concrete substrate cracks shrank as the yield point approached, and they propagated towards the neutral axis in the process. The SHCC layer was found to have multiple cracks in two different patterns. First, multiple cracks propagated arbitrarily all over the SHCC layer, as shown in Figure 11 as indicated by the blue dotted circle. Uniform tensile stress was induced in the layer due to beam bending, causing cracks to develop. 

In the second cracking pattern shown in Figure 11, multiple cracks had diffused from the concrete substrate. The substrate cracked due to diffuse cracking in the substrate that caused concentrated strain at the interface. During peak loading, the SHCC reached its ultimate strain capacity, resulting in crack localization. The SHCC layer and NC substrate displayed interfacial delamination at this stage of crack localization. Notably, this conclusion agrees with that published in [48]. The cause of delamination is the absence of steel across the interface of two layers, as the SHCC layer is only provided in the beam’s cover. The cracking mode of composite beam ‘RC-L-SHCC-60′ was similar to the previous one, even though the SHCC layer was reinforced in this specimen. The first visible crack was initiated in the concrete substrate at the mid-span of the beam at a load of around 21 kN (16.7% of the ultimate load). Cracks appeared in the SHCC and concrete substrate of the beam with an increase in load. Two cracking patterns (due to bending and strain concentration) were also observed in the SHCC layer.

For the composite beam RC-L-SHCC-90, the cracking pattern was slightly different from previous beams since half of its section was SHCC. Several microcracks occurred in the SHCC layer with the increase in loading. At the yielding point, very few cracks occurred in the concrete substrate, out of which only one concrete crack had fine crack diffusion in the SHCC layer. Also, several fine cracks randomly appeared in the SHCC layer, but they hardly propagated up to the center of the SHCC layer. At peak load, a crack in the only branched-out SHCC multiple cracking localized, and the SHCC layer ruptured.

#### 4.2.6. Crack Properties

The cracking pattern of beam specimen “RC-L-SHCC-60” has been enlarged, as shown in Figure 15. As the concrete crack width increased, SHCC cracks diffused into this specific location, forming a diffuse cracking pattern. The crack width in the strengthened beams was measured using image analysis software. There were less than 0.1 mm crack widths in the SHCC layer at the service stage and less than 1 mm crack widths at yielding. According to the ACI Committee Report 224R-01 [49], concrete cracks at the service stage should not be wider than 0.41 mm. As far as the crack widths were concerned, the SHCC-layered beams had a limited range of crack widths.

The RC beam crack widths, as determined at different loading stages (service, yielding, and ultimate), are listed in Table 8. In parenthesis, the crack width of each specimen is also compared with that of the control specimen. A maximum crack width of less than half was observed for SHCC-layered beams at the service stage, with RC-L-SHCC-90 having the lowest value. This trend can be observed in both the yielding and ultimate loading stages. As compared to beams with 30 mm SHCC layers, beams with 60 and 90 mm SHCC layers showed smaller crack widths. Conclusively, the application of SHCC layers in RC beams can limit crack width under service conditions.

Numbers and densities of cracks (number of cracks per mid-span length ‘550 mm’ of the beam) are presented in Table 9 for the concrete substrate and SHCC layer at the yield and ultimate phases. Throughout the entire loading period, six cracks were observed in the RC-C beam’s middle span. A similar number of cracks developed in the concrete substrate at the yielding stage, but an increase was observed at the ultimate stage for layered beams. It was the RC-L-SHCC-30 that had the most cracks in its SHCC layer out of all the layered beams. All of the layered beams had significantly more cracks in the SHCC layer than in the concrete substrate.

Furthermore, with increasing SHCC layer thickness, the crack density in the SHCC layer of the beams decreased. Overall, the crack control of RC-L-SHCC-30 and RC-L-SHCC-60 is better in terms of crack widths and densities. Since RC-L-SHCC-60 has improved the flexural performance over RC-L-SHCC-30, as discussed previously, it can be concluded that the SHCC layer of thickness 60 mm (30% of section height) is the most efficient layer thickness to be used in a composite beam to achieve better flexural and cracking performance.

## 5. Analytical Modeling

This study presents a modification of the modeling methodology recommended by [50]. The moment–curvature response of RC–SHCC-layered beams was predicted by an analytical model. In this study, a sectional analysis was performed using simplified material constitutive models as well as the equilibrium of internal forces and moments. Additionally, experimental results were compared with predictions to determine the accuracy of the proposed model.

### 5.1. Mechanical Behavior of Materials

#### 5.1.1. Steel

Using a bilinear constitutive model [46] for steel bars, the tensile constitutive model is simplified (Figure 16a and Equation (1)).
(1)$$\sigma_{s}=\left\{\begin{array}{cc} \varepsilon_{s}.E_{s}, & \varepsilon_{s}\leq \varepsilon_{sy}\\ f_{sy}, & \varepsilon_{sy}<\varepsilon_{s}\leq \varepsilon_{su} \end{array}\right\}$$
where
$\sigma_{s}$: the tensile stress in the steel bar.$\varepsilon_{s}$: the tensile strain in the steel bar.$f_{sy}$: the mean yielding stress of the steel bars (480 MPa).$\varepsilon_{sy}$: the strain corresponding to the yielding stress of the steel bars (0.24%).$E_{s}$: Young’s modulus of the steel bars (200 GPa).

#### 5.1.2. SHCC

The bilinear material model [28] for SHCC under tension is shown in Figure 7 and Equation (2).
(2)$$\sigma_{et}=\left\{\begin{array}{cc} \frac{f_{etc}}{\varepsilon_{etc}}.\varepsilon_{et}, & 0\leq \varepsilon_{et}\leq \varepsilon_{etc}\\ f_{etc}+\left(\frac{f_{etu}-f_{etc}}{\varepsilon_{etu}-\varepsilon_{etc}}\right).\left(\varepsilon_{et}-\varepsilon_{etc}\right), & \varepsilon_{etc}<\varepsilon_{et}\leq \varepsilon_{etu} \end{array}\right\}$$
where
$\sigma_{et}$: the tensile stress in the SHCC.$\varepsilon_{et}$: the tensile strain in the SHCC.$f_{etc}$: the tensile stress in the SHCC at first cracking (4 MPa).$\varepsilon_{etc}$: the tensile strain in the SHCC at first cracking (0.0326%).$f_{etu}$: the tensile stress in the SHCC at ultimate loading (4.35 MPa).$\varepsilon_{etu}$: the tensile strain in the SHCC at ultimate loading (2.0%).

As a point of clarification, in this study, the values of the *f_sy_*, *ε_sy_*, *f_etc_*, *ε_etc_*, *f_etu_* and *ε_etu_* are the average observed tensile properties of the steel and SHCC specimens.

#### 5.1.3. Concrete

In this study, the compressive stress–strain concrete model described in [46] was employed. A diagram of this model is shown in Figure 14a, which is derived from Equation (3).
(3)$$\begin{array}{c} \sigma_{c}=\left\{\begin{array}{cc} f_{c}\left(1-{\left(1-\frac{\varepsilon_{c}}{\varepsilon_{co}}\right)}^{n}\right), & 0\leq \varepsilon_{c}\leq \varepsilon_{co}\\ f_{c}, & \varepsilon_{co}<\varepsilon_{c}\leq \varepsilon_{cu} \end{array}\right\}\\ n=2-\frac{\left(f_{cu,k}-50\right)}{60}\\ \begin{array}{c} \varepsilon_{co}=0.002+0.5\left(f_{cu,k}-50\right)\times 10^{-5}\;\geq 0.002\;\\ \varepsilon_{cu}=0.0033-\left(f_{cu,k}-50\right)\times 10^{-5}\;\leq 0.0033 \end{array} \end{array}$$
where
$\sigma_{c}$: the compressive stress in the concrete.$\varepsilon_{c}$: the compressive strain in the concrete.$f_{c}$: the compressive strength of the concrete (30 MPa).$\varepsilon_{co}$: the strain at peak compressive stress of concrete (0.2%).$f_{cu,k}$: the cube compressive strength of the concrete (1.08% of the concrete’s cylinder compressive strength [51], 45 MPa).$\varepsilon_{cu}$: the strain at peak cube compressive stress of concrete (0.33%).n: the coefficient associated with the stress–strain response of concrete.

The values of *ε_co_*, *ε_cu_*_,_ and *n* are derived from Equation (3). The material model for the concrete under uniaxial tension is shown Equation (4).
(4)$$\begin{array}{c} \sigma_{ct}=\left\{\begin{array}{cc} \frac{f_{ctu}}{\varepsilon_{ctu}}\varepsilon_{ct}, & 0\leq \varepsilon_{ct}\leq \varepsilon_{ctu}\\ 0, & \varepsilon_{ct}>\varepsilon_{ctu} \end{array}\right\} \end{array}$$
where
$\sigma_{ct}$: the uniaxial tensile stress in the concrete.$\varepsilon_{ct}$: the uniaxial tensile strain in the concrete.$f_{ctu}$: the ultimate uniaxial tensile strength of the concrete (2.25 MPa).$\varepsilon_{ctu}$: the strain at peak ultimate uniaxial tensile stress of concrete (0.011%).

The values for *f_c_*, *f_ctu_* and *ε_ctu_* were selected in accordance with [46], corresponding to *f_cu,k_* and *f_ctu,k_*.

### 5.2. Methodology of the Sectional Analysis

A detailed analysis procedure for RC–SHCC composite beams was presented in [50]. A few assumptions were considered for the simplified analysis, i.e., plane sections remain plane under loading, no delamination at the substrate/SHCC interface, and there is a perfect bond between the matrix and rebars. The x-section’s stress–strain distributions for three loading stages (cracking, yielding, and peak) are shown in Figure 17, where *h_t_* is the thickness of the tensile region, *h_e_* is the SHCC layer thickness and *x* is the variable showing the height of any point from the bottom edge of the section. 

The basic force equilibrium equation (Ʃ*N* = 0) of a cross-section is shown in Equation (5), in which constitutive relationships of materials are substituted to obtain an equation to locate the neutral axis (*h_t_*) at different loading stages. The moment (*M*) at each stage can be calculated by the moment equilibrium equation (Ʃ*M* = 0) of the cross-section, as shown in Equation (6). In the case of SHCC-layered beams, cracking, yielding, and ultimate moments, as well as their corresponding curvatures (*ϕ*), were analyzed. The equations for *h_t_* and *M* for different loading stages are provided in the referred study [50] along with relevant checks. The composite beams’ curvature was calculated by dividing strain *ε_et_* with *h_t_* at different stages.
(5)$$\overset{h_{e}}{\underset{0}{\int }}\sigma \left(x\right)bdx+\overset{h_{t}}{\underset{h_{e}}{\int }}\sigma \left(x\right)bdx+\sigma_{s}A_{s}-\overset{h}{\underset{h_{t}}{\int }}\sigma \left(x\right)bdx=0$$
(6)$$M=\overset{h}{\underset{h_{t}}{\int }}\sigma \left(x\right)bxdx-\overset{h_{e}}{\underset{0}{\int }}\sigma \left(x\right)bxdx-\sigma_{s}A_{s}h_{s}-\overset{h_{t}}{\underset{h_{e}}{\int }}\sigma \left(x\right)bxdx$$

### 5.3. Prediction Performance of the Model

A comparison between the analytical and experimental moment–curvature relationships of RC–SHCC-layered beams is presented in Figure 18. It can be observed that both the predictive and the experimental curves exhibit similar behavior. With a higher SHCC layer thickness, the predictive curves showed increased elastic stiffness. In the analysis, the RC-L-SHCC-30 and RC-L-SHCC-60 exhibited cracking in the concrete first (before SHCC), and their analytical moments were lower than the experimental moments; it was vice versa in the case of the RC-L-SHCC-90.

The yield and peak moments of the layered beams predicted from the analysis are lower or equal to those of the experimental moments. With an increase in the SHCC layer thickness, the analytical yield and peak moment values of layered beams are found to be near the experimental values. The slight reduction in moments obtained from the theoretical analysis is understandable because the rebar yield strength was considered in this analysis, whereas in actuality, the post-yield capacity of rebar was utilized in the experimentation, as confirmed by the steel strains exceeding the yield values in the load–strain relationships. 

Moreover, the theoretical curvatures are also found to be lower than the experimental curvatures for all specimens. Since these calculated results are equal to or lower than the actual results, it can be observed that this procedure gives results on the conservative side of the analysis. The cracking, yielding, and ultimate moments and curvatures from the experimental and theoretical results are tabulated in Table 10 and Table 11, where subscripts ‘e’ and ‘a’ represent the experimental and analytical data of layered beams. Also, the ratios of the experimental and analytical results of M_a_/M_e_ are presented. 

Except for RC-L-SHCC-30, the percentage difference between analytical and experimental moments at all loading stages for RC-L-HCC-60 and RC-L-SHCC-90 is under 10%. The mean value of the specimen’s M_a_/M_e_ ratios at each loading stage is approximately 0.93. On the other hand, the mean value of the ratios of the experimental and analytical curvatures of the specimens at each loading stage ranges from 0.65 to 0.8. Therefore, it can be concluded that such a theoretical approach to section analysis using material models can be reasonably utilized to predict the flexural behavior of the RC–SHCC composite beams.

## 6. Conclusions, Limitations, and Prospective Studies

This study evaluated the effectiveness of strain-hardening cementitious composite layers in strengthening RC beams using experimental and analytical methods. It should be noted that these conclusions may only apply to precast concrete beams with layers of SHCC beams. Furthermore, the comparisons were made assuming that all beams had the same depth and rebar configuration. In light of this research, the following conclusions can be drawn:(1)SHCC mechanical properties vary within the limit of the reported variation range of 10 to 25%. Compared to the control RC beams, the RC-SHCC composite beams had a higher load-carrying capacity and stiffness due to the partial replacement of concrete with SHCCs. In comparison with normal beams, SHCC beams have a relatively insignificant improvement in flexural performance. It is evident from the cracking patterns of beams containing SHCC layers that these layers can control cracking by the multi-cracking response, thereby producing more durable structures. It was found that composite beams with an SHCC layer covering 30% of the beam height had higher maximum yield and ultimate loads than those with other SHCC layers. It was discovered that the specimens’ moment capacities followed similar trends.(2)As a result of the additional tensile area provided by the SHCC layer, RC-SHCC-layered beams had less ductility than the control beam. As a result of their higher ductility indexes than the minimum specified in FIP (1990), RC-SHCC-layered beams still provide an adequate warning before failure. According to the results of the layered beam, the SHCC layer of 30% of the beam height displayed the highest degree of ductility.(3)It was observed that the SHCC layers of the RC-SHCC beams had a substantial number of fine cracks (multiple cracking behaviors). A narrower crack width was also observed in the concrete substrate and the SHCC layer. Accordingly, the application of SHCC layers in RC beams can limit the crack width. The finding indicates that SHCC in beams can be utilized as a protective layer that controls crack widths, thereby improving their structural performance. The SHCC layer of thickness 60 mm (30% of section height) is the most efficient layer thickness to be used in a composite beam to achieve better flexural and cracking performance.(4)The stress–strain constitutive models of materials were used to calculate moments and curvatures at various loading stages. The predictions of the theoretical model were in fair agreement with the experimental data. The yield and peak moments of the layered beams predicted from the analysis are lower or equal to those of the experimental moments. The average value of the ratios between the experimental and analytical curvatures of the specimens at each loading stage varies between 0.65 and 0.8. It follows that it is reasonable to predict the flexural behavior of the RC-SHCC composite beams using a theoretical approach to section analysis employing material models.

Future studies might expand the current study by considering different concrete and/or SHCC depths, reinforcement ratios, and loading configurations. Additionally, the results of the current investigation may be reproduced with the use of different specimen replicas. A considerable margin of difference exceeding 5% was observed between the basic flexural test results in the present study. Thus, it is more likely that such discrepancies will be amplified by the size of the structural member.

## Figures and Tables

**Figure 1 materials-16-05135-f001:**
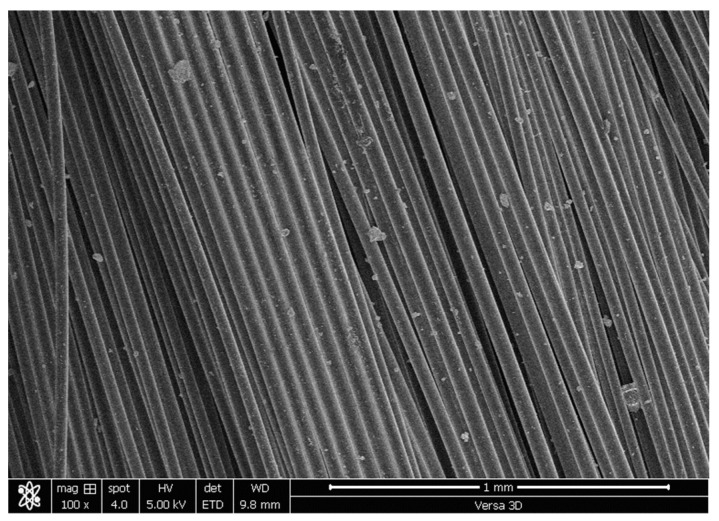
SEM image of PVA fiber strands.

**Figure 2 materials-16-05135-f002:**
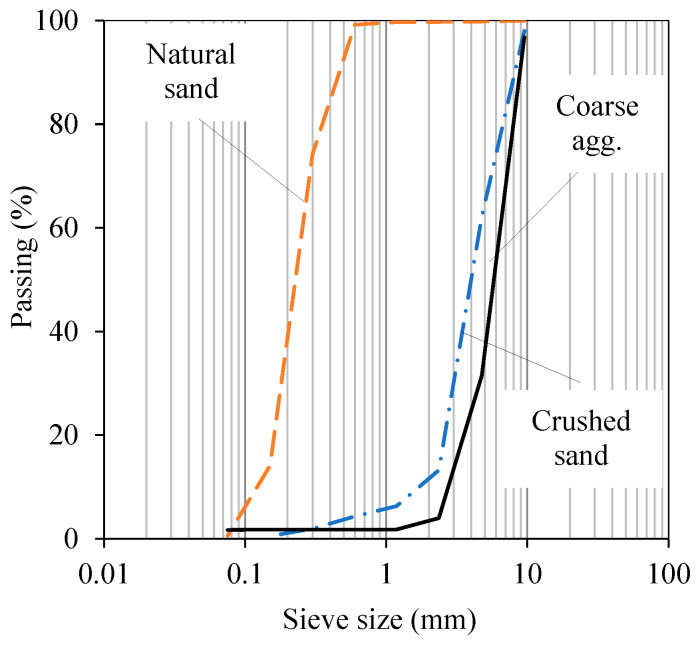
Sieve analysis of fine and coarse aggregates.

**Figure 3 materials-16-05135-f003:**
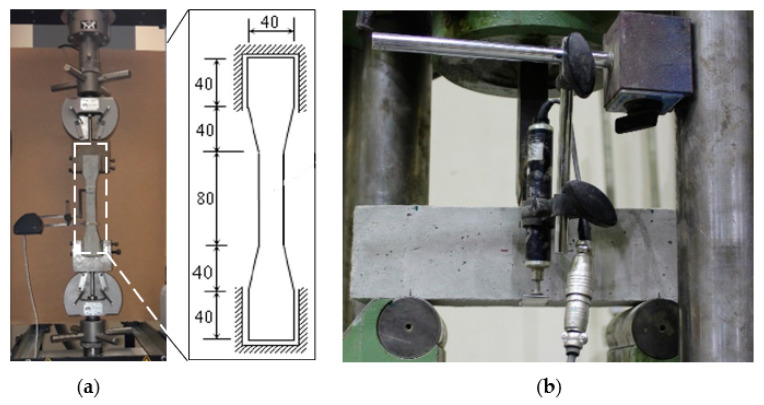
(**a**) Tension test and (**b**) flexural test setups for the SHCC samples.

**Figure 4 materials-16-05135-f004:**
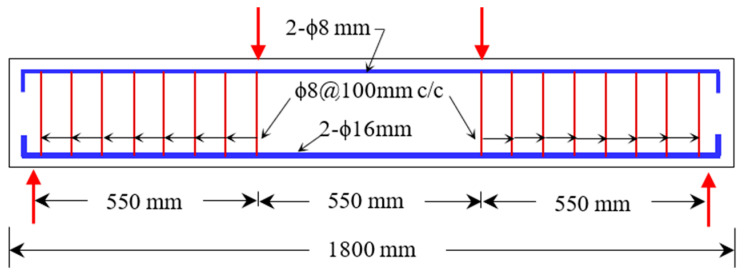
Rebar details of beam specimens.

**Figure 5 materials-16-05135-f005:**
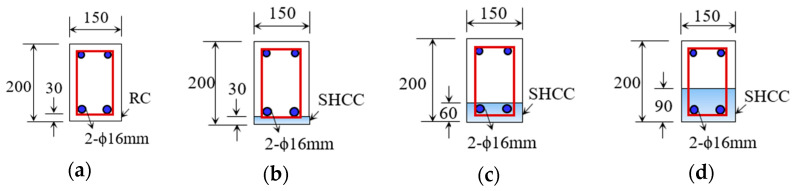
RC beam sections: (**a**) RC-C, (**b**) RC-L-SHCC-30, (**c**) RC-L-SHCC-60, and (**d**) RC-L-SHCC-90 (dimensions are in mm).

**Figure 6 materials-16-05135-f006:**
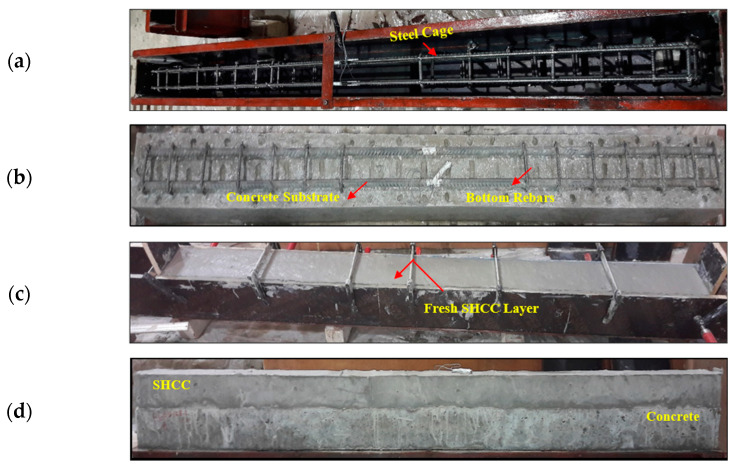
The casting process of beam specimens: (**a**) a steel cage, (**b**) an inverted hardened RC beam without a layer of SHCCs, (**c**) an inverted RC beam with a SHCC layer, and (**d**) a hardened RC-SHCC-layered beam.

**Figure 7 materials-16-05135-f007:**
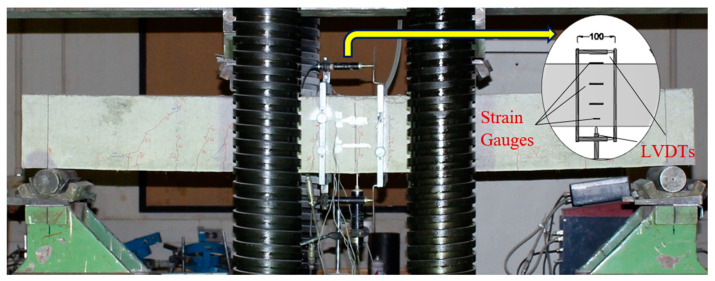
Test setup of specimens.

**Figure 8 materials-16-05135-f008:**
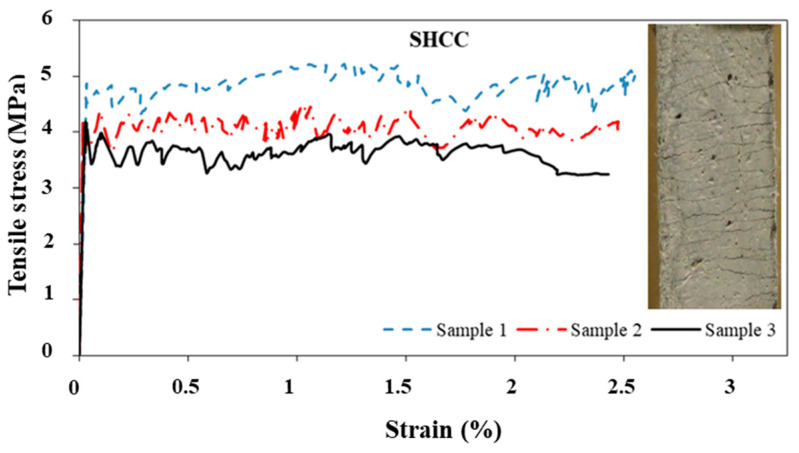
Response of the SHCC to uniaxial tension.

**Figure 9 materials-16-05135-f009:**
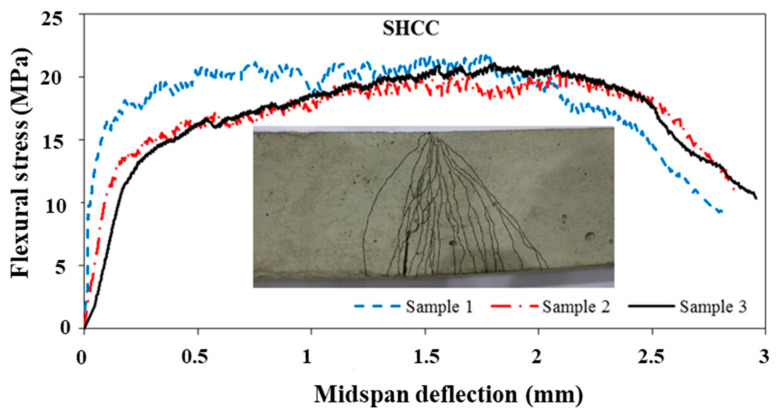
The properties of the SHCC under flexural loading.

**Figure 10 materials-16-05135-f010:**
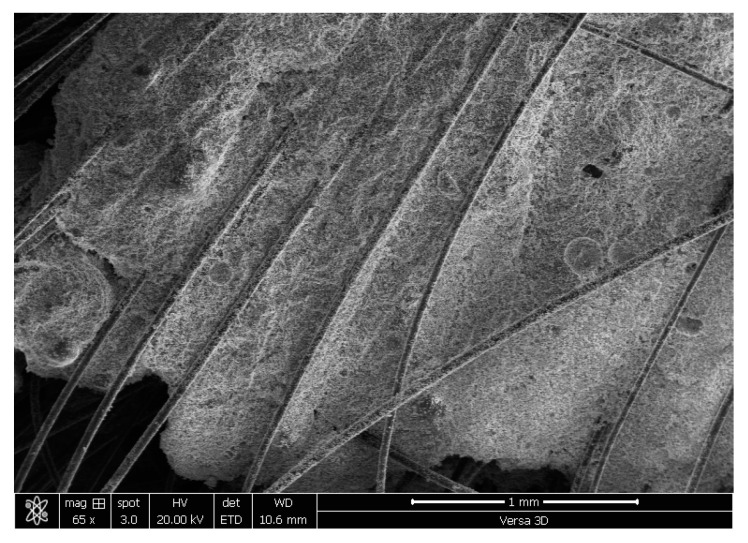
SEM image of the pieces extracted from specimens tested for flexural properties.

**Figure 11 materials-16-05135-f011:**
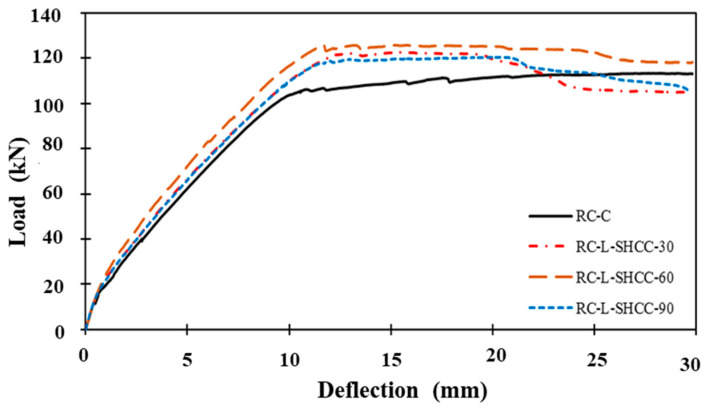
Load–deflection curves of beam specimens.

**Figure 12 materials-16-05135-f012:**
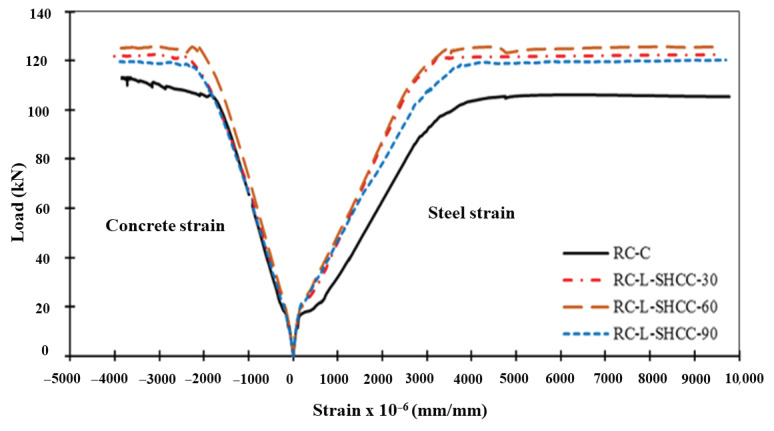
Load and strain curves of specimens.

**Figure 13 materials-16-05135-f013:**
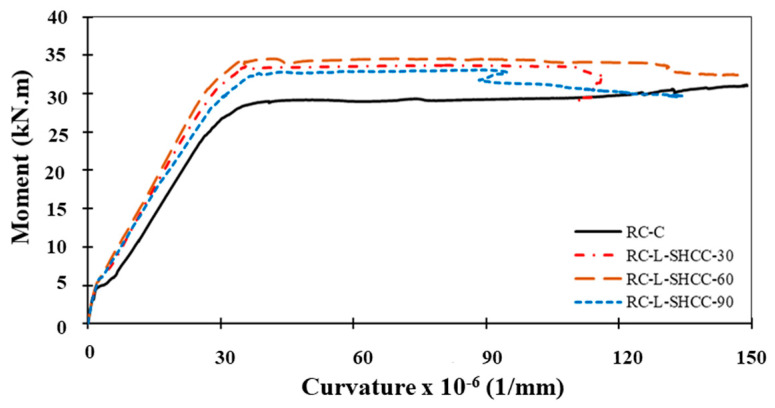
Moment–curvature curves of beam specimens.

**Figure 14 materials-16-05135-f014:**
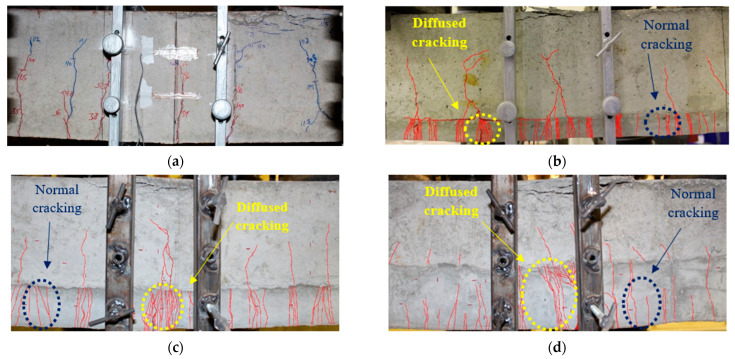
Cracking patterns of specimens: (**a**) RC-C, (**b**) RC-L-SHCC-30, (**c**) RC-L-SHCC-60, and (**d**) RC-L-SHCC-90.

**Figure 15 materials-16-05135-f015:**
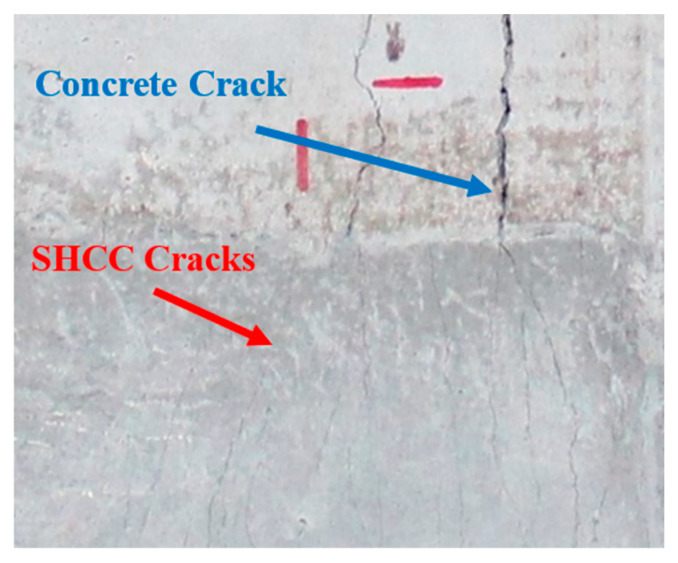
Magnified cracking in the ‘RC-L-SHCC-60’ beam.

**Figure 16 materials-16-05135-f016:**
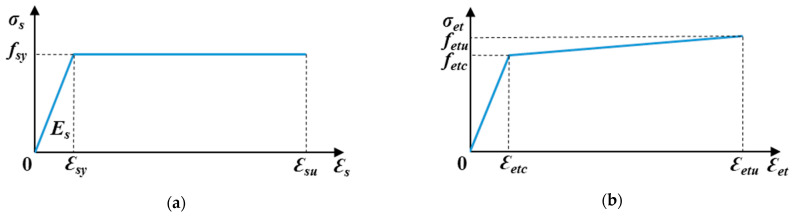
Idealized stress–strain behavior in tension. (**a**) Steel and (**b**) SHCC.

**Figure 17 materials-16-05135-f017:**
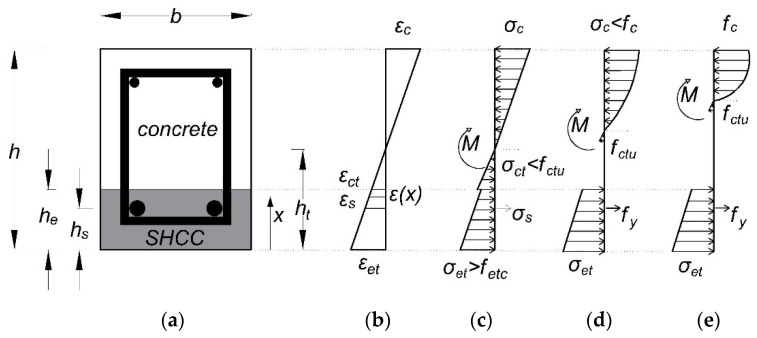
Properties of the layered RC beam: (**a**) cross-section; (**b**) strain profile; (**c**–**e**) stress profile at cracking, yielding, and ultimate stages, respectively.

**Figure 18 materials-16-05135-f018:**
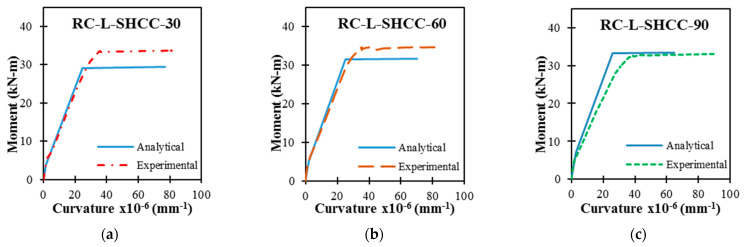
Prediction performance of the proposed analytical model. SHCC with (**a**) 30, (**b**) 60, and (**c**) 90 mm thickness.

**Table 1 materials-16-05135-t001:** Chemical analysis of cement and fly ash.

Oxides (%)	OPC	FA
SiO_2_	20.41	53.2
Al_2_O_3_	5.32	27.3
Fe_2_O_3_	4.1	4.03
CaO	64.14	0.90
MgO	0.71	0.60
SO_3_	2.44	0.20
TiO_2_	0.3	1.52
Na_2_O	0.10	0.22
K_2_O	0.17	1.22
L.O.I	2.18	10.02

**Table 2 materials-16-05135-t002:** Physicomechanical characteristics of the PVA fibers.

Length (mm)	Diameter (µm)	Specific Gravity	Tensile Strength (MPa)	Young’s Modulus (GPa)	Elongation (%)
12	40	1.3	1600	42	7

**Table 3 materials-16-05135-t003:** Mix design.

W/B	Unit Weight (kg/m^3^)				PVA Fiber (%, vol.)
	Cement	Fly Ash	Dune Sand	Water	
0.26	555	666	466	315	2

**Table 4 materials-16-05135-t004:** Mixture proportions of concrete.

W/C	Unit Weight (kg/m^3^)				
	Cement	Coarse Aggregate	Fine Sand	Crushed Sand	Water
0.5	350	1070	488	262	175

**Table 5 materials-16-05135-t005:** Critical load–deflection values.

Beam	Elasticity ^a^		Yielding ^b^		Peak ^c^		Ultimate ^d^	
	Load	Deflection	Load	Deflection	Load	Deflection	Load	Deflection
	kN	mm	kN	mm	kN	mm	kN	mm
RC-C	15.25	0.62	86.61	7.60	113.17	34.32	90.45	40.94
RC-L-SHCC-30	20.16	0.79	112.00	10.33	122.57	15.52	97.12	37.69
RC-L-SHCC-60	21.45	0.82	113.27	9.52	125.44	17.27	98.81	42.75
RC-L-SHCC-90	21.82	0.97	102.54	9.06	120.30	20.28	96.00	31.04

^a^ first-cracking stage, ^b^ tensile rebar yielding stage, ^c^ peak load stage, ^d^ ultimate failure stage.

**Table 6 materials-16-05135-t006:** Critical moment–curvature values.

Beam	Elasticity		Yielding		Ultimate	
	Moment	Curvature	Moment	Curvature	Moment	Curvature
	kNm	µrad	kNm	µrad	kNm	µrad
RC-C	4.2	1.6	28.6	36.4	31.1	148.8
RC-L-SHCC-30	5.5 (31%)	2.2 (38%)	33.3 (16%)	34.6 (−5%)	33.4 (7%)	109.5 (−26%)
RC-L-SHCC-60	5.9 (40%)	2.7 (69%)	34.0 (19%)	33.7 (−8%)	34.0 (9%)	126.7 (−15%)
RC-L-SHCC-90	6.0 (43%)	3.0 (88%)	32.3 (13%)	36.6 (0%)	33.0 (6%)	91.6 (−38%)

**Table 7 materials-16-05135-t007:** Ductility indexes of the RC beams.

Beam	Curvature ϕ		Deflection δ		Section Ductility	Member Ductility
	Yield	Ultimate	Yield	Ultimate	µ_sec_ = ϕ_u_/ϕ_y_	µ_mem_ = δ_u_/δ_y_
	ϕ_y_ (10^−6^/mm)	ϕ_u_ (10^−6^/mm)	δ_y_ (mm)	δ_u_ (mm)		
RC-C	36.4	148.8	10.2	29.8	4.1	2.93
RC-L-SHCC-30	34.6	109.5	12.1	19.4	3.2 (−22%)	1.61 (−45%)
RC-L-SHCC-60	33.7	126.7	11.2	24.1	3.8 (−8%)	2.14 (−27%)
RC-L-SHCC-90	36.6	91.6	11.4	21.0	2.5 (−39%)	1.83 (−37%)

**Table 8 materials-16-05135-t008:** Maximum concrete crack width at different loading stages.

Beam	At Service Load (mm)	At Yielding Load (mm)	At Ultimate Load (mm)
RC-C	3	5.8	13
RC-L-SHCC-30	1.5 (−50%)	3.5 (−40%)	6 (−54%)
RC-L-SHCC-60	0.5 (−84%)	2.2 (−62%)	6 (−54%)
RC-L-SHCC-90	0.4 (−87%)	1.8 (−69%)	5.8 (−55%)

**Table 9 materials-16-05135-t009:** Cracks in the mid-span of the specimens.

Beam	Cracks at the Yielding Stage				Cracks at the Ultimate Stage			
	Concrete		SHCC		Concrete		SHCC	
	Number (#)	Density (#/mm)	Number (#)	Density (#/mm)	Number (#)	Density (#/mm)	Number (#)	Density (#/mm)
RC-C	6	0.012	-	-	6	0.012	-	-
RC-L-SHCC-30	6	0.012	37	0.074	8	0.016	80	0.16
RC-L-SHCC-60	7	0.014	31	0.06	9	0.018	55	0.11
RC-L-SHCC-90	6	0.012	25	0.05	9	0.018	42	0.084

**Table 10 materials-16-05135-t010:** Comparison of cracking, yield, and the ultimate moment of layered beams (kN-m).

Beam	Experimental			Analytical			Ratio		
	M_cr,e_	M_y,e_	M_u,e_	M_cr,a_	M_y,a_	M_u,a_	M_cr,a_/M_cr,e_	M_y,a_/M_y,e_	M_u,a_/M_u,e_
RC-L-SHCC-30	5.66	33.27	33.5	3.74	29.05	29.36	0.66	0.87	0.88
RC-L-SHCC-60	5.45	34.51	34	5.17	31.46	31.69	0.95	0.91	0.93
RC-L-SHCC-90	5.94	32.27	33	7.09	33.16	33.35	1.19	1.03	1.01
Mean							0.93	0.93	0.94

**Table 11 materials-16-05135-t011:** Comparison of cracking, yield, and the ultimate curvature of layered beams (10^−6^ mm^−1^).

Beam	Experimental			Analytical			Ratio		
	φ_cr,e_	φ_y,e_	φ_u,e_	φ_cr,a_	φ_y,a_	φ_u,a_	φ_cr,a/_φ_cr,e_	φ_y,a/_φ_y,e_	φ_u,a/_φ_u,e_
RC-L-SHCC-30	2.37	34.58	109.55	1.318	24.5	74.6	0.56	0.71	0.68
RC-L-SHCC-60	2.22	35.23	126.74	1.957	25.1	68.6	0.88	0.71	0.54
RC-L-SHCC-90	2.83	36.60	91.63	2.767	25.8	63.6	0.98	0.71	0.69
Mean							0.81	0.71	0.64

## Data Availability

Data is contained within the article.

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
