# Peer review of "Structural Performance Improvement of Reinforced Concrete Beams by Strain-Hardening Cementitious Composite Layers"

_materials, 2023, doi:10.3390/ma16145135_

Round 1
Reviewer 1 Report
The issue presented in the paper is interesting and certainly worth investigating. However, in its present form, I found the analysis presented in the paper insufficient to be published.
The number of specimens tested in a single configuration is inadequate. A minimum of three specimens should be tested to derive meaningful conclusions, particularly when analysing any fibre reinforced concrete. Meanwhile, the main tests, where the reinforced beams were partially composed of ordinary concrete and partially of SHCC, were performed only on one specimen. The results could vary significantly when more of these complex elements are tested. The basic tests suggest this possibility. In this study, only three specimens underwent basic compressive, flexural, and tensile strength tests. The authors observed variations among the results, as depicted in Figure 6 and Figure 7. A different mechanical response in the elastic stage was observed in Figure 7. The stress-deflection diagram's progression should be comparable in this stage for the same material. The fibres begin to function after reaching the tensile strength, and during this phase, the response may differ depending on the number and orientation of the fibres in the cross-section. These results need to be thoroughly reviewed initially.
Moreover, the paper seems to be prepared with no care, please see some comments:
· Point 2 – some error after the Table2.
· The points 5.1.1 and 5.1.2 are missing.
· The indexes were not changed in the equation (4).
Author Response
Response to Reviewer #1 Comments
General comment. The issue presented in the paper is interesting and certainly worth investigating. However, in its present form, I found the analysis presented in the paper insufficient to be published.
Response. We are deeply grateful to the honorable reviewer for his encouraging, detailed, and constructive comments. We have included all comments of the honorable reviewer in the revised manuscript. Our pointwise response to the comments is as follows.
Specific comments.
Comment#1. The number of specimens tested in a single configuration is inadequate. A minimum of three specimens should be tested to derive meaningful conclusions, particularly when analysing any fibre reinforced concrete. Meanwhile, the main tests, where the reinforced beams were partially composed of ordinary concrete and partially of SHCC, were performed only on one specimen. The results could vary significantly when more of these complex elements are tested. The basic tests suggest this possibility. In this study, only three specimens underwent basic compressive, flexural, and tensile strength tests. The authors observed variations among the results, as depicted in Figure 6 and Figure 7. A different mechanical response in the elastic stage was observed in Figure 7. The stress-deflection diagram's progression should be comparable in this stage for the same material. The fibres begin to function after reaching the tensile strength, and during this phase, the response may differ depending on the number and orientation of the fibres in the cross-section. These results need to be thoroughly reviewed initially.
Response. We would like to express our appreciation to the esteemed reviewer for his insightful comments. The variation within the specimens was within the commonly accepted range for the flexural test and yet for the tensile test, it is expected as it depends on the gripping system which cannot be fully controlled. However, in the flexural analysis, which is the main theme of the work, where the setup variation is minimal, it can be seen that the variation in the results is also limited and within the acceptable 5% error. Based on these facts, a structural beam specimen for each test was prepared and tested. With respect to the fiber orientation, the authors have published different work showing that the optimized mix composition delivers the optimum fiber dispersion regardless of the specimen dimension and accordingly a comparable response is expected:
M.I. Khan, G. Fares, S. Mourad, W. Abbass. 2016. Optimized fresh and hardened properties of SHCC: Effect of sand size and workability. Journal of Materials in Civil Engineering, Vol. 28, No. 12. http://dx.doi.org/10.1061/(ASCE)MT.1943-5533.0001665.
M.I. Khan, S. Mourad, G. Fares, “Effect of different natural sands on the tensile behaviour of SHCC under the effect of different strain rates”. The Eighth International Structural Engineering and Construction Conference. Sydney, Australia, November 23-28, 2015.
M.I Khan and G. Fares, SHCC Design and Manufacturing Challenge Using Non-Standard Materials, Proceedings of the First International Symposium on Innovations in Civil Engineering and Technology, I CIVILTECH 2019, 23-25 October 2019, pp 333-340, Afyonkarahisar, Turkey.
Comment#2. Moreover, the paper seems to be prepared with no care, please see some comments: Point 2 – some error after the Table 2.
Response. Thank you for drawing our attention to these errors. A number of amendments and corrections have been made as a result.
Comment#3. The points 5.1.1 and 5.1.2 are missing.
Response. Sections are 5.1.1 and 5.1.2 are for the steel and SHCC materials models. These sections are available in the revised manuscript.
Comment#4. The indexes were not changed in the equation (4).
Response. Thank you for drawing our attention to this typo error. It has been addressed in the revised manuscript.
Reviewer 2 Report
Dear Authors,
The research results are very interesting and deserve publication. However, You should complete the following information:
1. check the chapter - introduction and literature review again.
2. complete the information in the 'Materials' chapter. Coarse aggregate properties are missing. In general, the screening curves of all aggregates should be presented on the graph.
3. why should the density of cement and ash be of interest? Do they stand out from the standard features? Please specify the chemical composition of cement and ash.
4. the Blein specific surface area is given in cm2/g
5. please show the fibers used in the picture.
Reviewer
Author Response
Response to Reviewer #2 Comments
General comment. Dear Authors, the research results are very interesting and deserve publication. However, you should complete the following information:
Response. We would like to express our appreciation to the esteemed reviewer for his encouraging comments. Here are our point-by-point responses to the comments.
Specific comments.
Comment#1. Check the chapter - introduction and literature review again.
Response. Done as requested.
Comment#2. Complete the information in the 'Materials' chapter. Coarse aggregate properties are missing. In general, the screening curves of all aggregates should be presented on the graph.
Response. Added as requested.
Comment#3. Why should the density of cement and ash be of interest? Do they stand out from the standard features? Please specify the chemical composition of cement and ash.
Response. Actually, it is a normal interest as we use in the mix design. The properties of cement and fly ash are added to the revised manuscript. Please check Table 1/
Comment#4. The Blein specific surface area is given in cm2/g
Response. Done as requested.
Comment#5. Please show the fibers used in the picture.
Response. Thank you for bringing this important note to our attention. In Figure 1, we have presented the fibers we used.
Reviewer 3 Report
The paper is interesting and can be accepted for publication after several revisions suggested by this reviewer:
1. Introduction:
- Too many short paragraphs, please combine them.
- The limitations of previous literature should be clarified, and the new contribution of the present study should be emphasized.
2. Materials:
Why NC was prepared with coarse aggregates no larger than 10 mm in size? Please clarify.
3. Methods:
The tensile setup of the SHCC is performed based on what standard or study? Please clarify.
4. The test parameters of this study are not clear. Please significantly revise.
5. From the test results, the performance improvement of beams with SHCC is not significant compared to normal beams. Please discuss this issue.
Author Response
Response to Reviewer #3 Comments
General comment. The paper is interesting and can be accepted for publication after several revisions suggested by this reviewer.
Response. The authors would like to thank the esteemed reviewer for the encouraging comments he made about the paper. Our response to the comments has been outlined in the following points-by-points.
Specific comments.
Comment#1. Introduction: Too many short paragraphs, please combine them.
Response. Thank you for drawing our attention to this matter. A number of amendments and corrections have been made as a result.
Comment#2. The limitations of previous literature should be clarified, and the new contribution of the present study should be emphasized.
Response. Done as requested.
Comment#3. Materials: Why NC was prepared with coarse aggregates no larger than 10 mm in size? Please clarify.
Response. A coarse aggregate of a nominal size of 10 mm was used in the preparation of the concrete. The revised version of the paper has addressed this error in the description in order to make it more accurate.
Comment#4. Methods: The tensile setup of the SHCC is performed based on what standard or study? Please clarify.
Response. Additional details regarding this test are provided in the below references, a collection of the authors' previous research studies. This clarification has been added to the revised manuscript.
M.I. Khan, S. Mourad, G. Fares, “Effect of different natural sands on the tensile behaviour of SHCC under the effect of different strain rates”. The Eighth International Structural Engineering and Construction Conference. Sydney, Australia, November 23-28, 2015.
M.I. Khan, S. Mourad, G. Fares. Strain capacity of SHCC using different strain loading rates. Advances in Construction Materials Proceedings of the ConMat’15 Conference, the Fifth International Conference on Construction Materials: Performance, Innovations and Structural Implications, 19-21, August 2015, Eds. N. Banthia, T. Miyagawa, C. Miao, K. Kobayashi, Whistler, The University of British Columbia, BC, Vancouver, Canada, pp. 1605-16014.
Fares, G., Khan, M.I., Mourad, S. and Abbass, W., 2015. Evaluation of PVA and PBI-based engineered-cementitious composites under different environments. Construction and Building Materials, 85, pp.109-118.
Comment#5. The test parameters of this study are not clear. Please significantly revise. compared to normal beams. Please discuss this issue.
Response. Thank you for bringing this to our attention. We have amended the last paragraph of the introduction to address this concern.
Comment#6. From the test results, the performance improvement of beams with SHCC is not significant compared to normal beams. Please discuss this issue.
Response. Thank you for drawing our attention to this issue. A discussion on this issue has been incorporated into the revised manuscript's conclusions in response to this comment.
Round 2
Reviewer 1 Report
Dear Authors,
I regret to write you that the manuscript has not undergone revision in accordance with the initial review. The inclusion of an adequate number of specimens is indispensable to attain reliable and significant test results. The magnitude of variation observed in the outcomes of basic flexural tests exceeded 5% by a substantial margin. Consequently, it is reasonable to expect that such discrepancies would be further amplified when applied to larger structural beams.
Author Response
Response to Reviewer #1 Comments
Comment#1. Are the results clearly presented? Must be improved.
Response. We are grateful for the help provided by the honorable reviewer in disseminating the results in the best possible way. Our presentation of the results has been improved in response to his valuable feedback.
Comment#2. Are the conclusions supported by the results? Must be improved.
Response. The conclusion section has been enhanced as per the valuable comment.
Comment#3. I regret to write you that the manuscript has not undergone revision in accordance with the initial review. The inclusion of an adequate number of specimens is indispensable to attain reliable and significant test results. The magnitude of variation observed in the outcomes of basic flexural tests exceeded 5% by a substantial margin. Consequently, it is reasonable to expect that such discrepancies would be further amplified when applied to larger structural beams.
Response. The logistic limitations preclude addressing this concern in the current research. Accordingly, it would be handled in a future study, as indicated in section 6 of the revised manuscript.
Reviewer 3 Report
The paper is now acceptable for publication.
Author Response
Response to Reviewer #2 Comments
General comment. The paper is now acceptable for publication.
Response. We are grateful to the honorable reviewer for his decision; the amount of time and effort he put into this review is much appreciated.